# Global variation of low bone mineral density in special olympics adult athletes with intellectual and developmental disability—A cross-sectional study

**Mary Pittaway**[ID][1], **John P. Hanley**[2], **Andrew E. Lincoln**[ID][2], **Alicia M. Dixon-Ibarra**[ID][2], **John T. Foley**[ID][3]*

**1** Diversified Resources, Missoula, Montana, United States of America, **2** Special Olympics International, Washington, Columbia, United States of America, **3** Department of Physical Education, State University of New York at Cortland, Cortland, New York, United States of America

* John.foley@cortland.edu

## Abstract

Adults with intellectual and developmental disabilities (IDD) face a high risk of low bone mineral density (LBMD), a key osteoporosis indicator, yet global data remains limited. Understanding LBMD prevalence among adults with IDD is crucial for targeted public health interventions. This study examines LBMD variations in Special Olympics athletes, stratified by age, sex, and World Health Organization (WHO) global region, and explores bone mineral density (BMD) levels achieved during peak bone mass (PBM) age (20–29 years). This cross-sectional study analyzed data from 25,868 Special Olympics athletes (20+ years) screened in Healthy Athletes between 2011–2023. BMD testing was conducted by licensed clinicians, and institutional review board approval was obtained to use this deidentified data. BMD was assessed via Quantitative Ultrasound, and WHO criteria classified T-score status. LBMD prevalence (<-1.0 T-score) was examined across age, sex, and global regions. Chi-square, rate ratios with 95% confidence intervals, and binomial logistic regression were calculated to analyze LBMD and PBM across age, WHO regions, and sex. Overall, 26.9% had LBMD, with similar prevalence in males (27.3%) and females (26.3%). LBMD increased annually by 1.43% in males and 2.50% in females. Highest LBMD prevalence was seen in Eastern Mediterranean females (52.4%) and Southeast Asian males (48.7%) and females (45.5%). Alarmingly, 54.9% failed to achieve optimal BMD (≥0.0) before age 30, with 24.4% of 20–29-year-olds already having LBMD. These findings highlight the need for early interventions, including nutrition, weight-bearing exercise, and routine screening, to improve bone health and reduce healthcare costs in adults with IDD. Policymakers must prioritize bone health initiatives to address disparities and enhance lifelong skeletal health.

**Data availability statement:** Researchers can request data from Special Olympics International at: https://resources.specialolympics.org/health/requesting-special-olympics-data. Additionally, we have uploaded data in the form of Supporting Information file (S1 Data. csv).

**Funding:** This study was financially supported by the Centers for Disease Control and Prevention (CDC) of the United States Department of Health and Human Services (HHS) (https://www.cdc.gov) in the form of a financial assistance award (NU27DD000021). No additional external funding was received for this study. The contents of this study are those of the authors, and do not necessarily represent the official views of, nor an endorsement, by CDC/HHS, or the U.S. Government.

**Competing interests:** The authors have declared that no competing interests exist.

## Introduction

Bone health is an important part of a person's physical health and should be considered throughout the lifespan. Bone cells continuously turn over due to bone-forming osteoblasts and bone-resorbing osteoclasts [1]. When bone health is not prioritized, the imbalance in the turnover of bone cells can result in negative health outcomes, such as osteoporosis. This common condition of aging, where bone mineral density (BMD) is reduced and bone microarchitecture deteriorates, increases the risk of fracture, mortality, and financial burden [2]. Despite well-documented risks, population-level data on BMD in people with intellectual and developmental disabilities (IDD) remain limited. This study aims to address this gap by examining the trajectory of quantitative ultrasound (QUS) BMD T-scores from early adulthood onward, to determine when and how these scores begin to diverge from those observed in the general population.

Attaining peak bone mass (PBM), the greatest amount of bone an individual can acquire, is key to preventing osteoporosis and related fractures later in life [3,4] Depending on the skeletal site, PBM occurs between 20–29 years old [5]. In addition to genetics and certain medical conditions, low PBM or the failure to accrue a healthy PBM (QUS T-score of ≥ 0.0) is affected by nutrition, vitamin D levels, physical activity, hormone status, medications and alcohol and/or tobacco use pre-adulthood [6–9]. Although bone mass typically declines with age, children and adolescents who develop higher PBM reduce their risk of fractures and osteoporosis later in life. Males typically achieve higher PBM than females and this increased BMD tracks throughout life until about age 70 with hormonal decline [10]. While attainment of PBM is important for everyone, this issue is particularly crucial for people with IDD, who face an increased risk of early onset osteoporosis, decreased life expectancy, and pronounced increases in morbidity, partially due to premature onset of age-related conditions [11,12]. Despite well-documented risks, population-level data on BMD in individuals with IDD remain limited. This study aims to address this gap by examining the trajectory of quantitative ultrasound (QUS) BMD T-scores from early adulthood onward, to determine when and how these scores begin to diverge from those observed in the general population.

IDD is defined as having an IQ below 70, adaptive impairments that affect an individual's growth and well-being, and onset prior to age 22 [13]. According to the World Health Organization (WHO), up to three percent of people worldwide have IDD, constituting the world's largest disability population.

Adults with IDD are at an increased risk of low bone mineral density (LBMD) due to a combination of non-modifiable, modifiable, and social risk factors [11,14]. As a result, they experience higher rates of osteoporosis and fractures—up to three times the rate of the general population [15]. The age of onset for major osteoporotic fractures and hip fractures occurs approximately 15 and 20 years earlier, respectively, in women with IDD compared to those without IDD, and 20 and 30 years earlier, respectively, in men [16]. Moreover, the distribution of fractures in adults with IDD differs markedly from that in the general population, with hip fractures accounting for a higher proportion of all fractures in those with IDD. The incidence, type, and

distribution of fractures in this population strongly suggest the presence of early-onset osteoporosis [15]. Current knowledge of BMD in adults with IDD remains limited, particularly in terms of insights from global, large-scale samples that could identify optimal intervention points across different regions and stages of life.

Special Olympics is the largest sport organization for people with IDD, serving over 3.5 million athletes worldwide. Leveraging sport as a platform, Special Olympics provides health programming designed to enhance the health and well-being of people with IDD. Special Olympics Healthy Athletes (HA) offers free health screenings, education, and follow-up referrals to adults with IDD who participate in local, national, and international events. Since 1997, Special Olympics has performed over two million screenings across eight health disciplines for athletes in nearly every country. BMD testing is a component of the Health Promotion discipline within the global Healthy Athletes program. Using this data allows researchers to expand the knowledge of the health of adults with IDD [17]. The purpose of this study was to identify the overall prevalence of LBMD among adults with IDD who participate in Special Olympics ("athletes") and which groups were at greatest risk of poor bone health in terms of age groups, sex, and WHO global region. A secondary aim was to determine the prevalence of PBM of participants at 20–29 years of age.

## Methods

### Procedures

Secondary data for this study were obtained from the 2011–2023 Special Olympics Healthy Athletes database. All athletes and/or their legal guardians provided consent for their HA screening data to be used for research purposes. Institutional Research Board Approval was obtained from SUNY Cortland for secondary data analysis (IRB approval: 151618). BMD testing was conducted at HA screenings by licensed clinicians or clinicians-in-training. Quality control procedures were part of clinician training provided prior to BMD testing with athletes [18,19]. BMD QUS T-scores were obtained using the Hologic Sahara Bone Densitometer, the Osteosys Sonost 3000, or the GE Achilles Express 3000. These ultrasound devices measure broadband ultrasound attenuation (BUA) and speed of sound (SOS) to determine QUS T-scores. Calcaneal QUS is a radiation-free alternative to dual-energy x-ray absorptiometry (DXA) to assess BMD [20]. Special Olympics Health Screenings are performed by clinical volunteers who follow Special Olympics guidelines. Quality Control (QC) procedures, including routine calibration as outlined in the manufacturer's operator manuals and volunteer training protocols, were followed daily prior to conducting BMD testing with athletes, to ensure the delivery of reliable, accurate, and consistent results. These practices included daily calibration with phantoms, image quality checks, reproducibility testing, regular maintenance, and adherence to regulatory standards. The Osteosys, Achilles Express and Hologic Sahara may be used for screening or monitoring, but T-scores from these devices might be considered supplementary and not diagnostic in the same way as DXA-based T-scores [21]. QUS showed a pooled sensitivity of ~80% and specificity of ~65% when using T-score cutoffs comparable to DXA and is effective at identifying individuals at high risk of low bone density [22,23]. Therefore, QUS is a useful screening tool for identifying individuals at risk of low bone mineral density (LBMD) and aids in estimating the proportion of a population needing further osteoporosis risk evaluation [24]. To identify individuals with LBMD, device-specific thresholds defined for QUS were followed [25]. To account for gait abnormalities, commonly observed in people with IDD, both heels were tested with the lowest of the two QUS T-scores recorded [26].

### Statistical analysis

LBMD is defined as any bone density classified as either osteoporosis or osteopenia. BMD QUS T-scores were classified using WHO guidelines where osteoporosis is defined as T-scores [-4.0, −2.5], osteopenia is (-2.5, -1.0), normal BMD is [-1.0, 1.0], and high normal BMD is (1.0, 5.0). Rates of LBMD were stratified across age groups, sex, and WHO regions.

When comparing LBMD rates across WHO regions, results were age-normalized by sex to the Americas region, so that each region would have the same age structure. When comparing LBMD prevalence rates within WHO regions, females were age-normalized to match the age distribution of the males in their region. In order to normalize the age data, ages

were grouped in ten-year intervals. LBMD prevalence rates were calculated with margins of error at a 95% confidence level (CIs). Chi-square and prevalence rate ratios with 95% CIs were calculated. Both crude and adjusted binomial logistic regressions were performed for females and males to measure the increase in LBMD with age as a continuous variable and WHO region as a confounding variable. To examine PBM, data was restricted to individuals aged 20–29 years. Data were analyzed using R version 4.3.1.

## Results

### Demographics

Between 2011–2023, there were 26,152 athlete screenings where athletes received BMD testing. Only athlete screenings where the athlete's age was between 20 and 89 years old, had a biological sex of female or male, and had the country of the athlete were used in this analysis. Therefore, this study focuses on 25,868 athlete screenings and represents Special Olympics athletes from 179 countries who received BMD testing. Athletes' median age was 30 years (range: 20–85), 58.6% were male, and 68.4% were from the Americas region (Table 1).

### Low bone mineral density prevalence

The overall prevalence of LBMD was 26.9% (26.3%-27.4%) with the prevalence increasing with age for both females and males (Fig 1, Table 2, S1 Fig).

### Low bone mineral density - age

The crude binomial logistic regression model revealed that for every year of age, the prevalence of LBMD increased 1.43% [1.11%, 1.76%] (Z = 8.70, $p < .0001$) for males and 2.50% [2.12%, 2.89%] (Z = 12.83, $p < .0001$) for females. When

**Table 1. Demographics and global regional distributions of Special Olympics athletes[†].**

| Characteristics | Africa | Americas | Eastern Mediterranean | Europe | Southeast Asia | Western Pacific | Total |
|---|---|---|---|---|---|---|---|
| **Age[*]** | | | | | | | |
| 20-29 | 361 (77.6%) | 7685 (43.4%) | 409 (71.0%) | 2267 (50.0%) | 747 (82.6%) | 1124 (66.9%) | **12,593 (48.7%)** |
| 30-39 | 75 (16.1%) | 5236 (29.6%) | 141 (24.5%) | 1342 (29.6%) | 122 (13.5%) | 374 (22.3%) | **7,290 (28.2%)** |
| 40-49 | 21 (4.5%) | 2767 (15.6%) | 23 (4.0%) | 641 (14.1%) | 25 (2.8%) | 119 (7.1%) | **3,596 (13.9%)** |
| 50-59 | 7 (1.5%) | 1,499 (8.5%) | 3 (0.5%) | 228 (5.0%) | 5 (0.6%) | 50 (3.0%) | **1,792 (6.9%)** |
| 60+ | 1 (0.2%) | 519 (2.9%) | 0 (0.0%) | 60 (1.3%) | 5 (0.6%) | 12 (0.7%) | **597 (2.3%)** |
| **Sex[**]** | | | | | | | |
| Female | 176 (37.8%) | 7,618 (43.0%) | 198 (34.4%) | 1,785 (39.3%) | 337 (37.3%) | 596 (35.5%) | **10,710 (41.4%)** |
| Male | 289 (62.2%) | 10,088 (57.0%) | 378 (65.6%) | 2753 (60.7%) | 567 (62.7%) | 1,083 (64.5%) | **15,158 (58.6%)** |
| **Total** | **465 (1.8%)** | **17,706 (68.4%)** | **576 (2.2%)** | **4,538 (17.5%)** | **904 (3.5%)** | **1,679 (6.5%)** | **25,868** |

[†]All data are derived from Special Olympics athletes ≥20 years old in the Special Olympics Healthy Athletes database.

Note: Pearson Chi-Square test on:

[*]Age and WHO regions: X-squared = 1278.4, df = 20, p-value < 0.001;

[**]Sex and WHO regions: X-squared = 71.825, df = 5, p-value < 0.001.

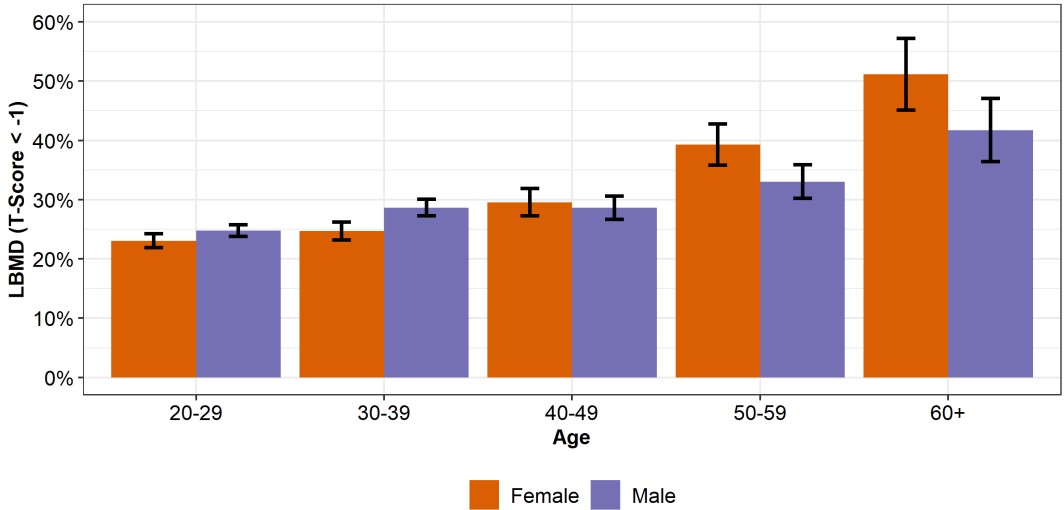

**Fig 1. Prevalence rates of LBMD for males and females across age groups and sex.** Error bars represent 95% confidence intervals.

**Table 2. Prevalence rates and prevalence rate ratios of LBMD by age group and sex with 95% confidence intervals. In addition, the Chi-squared statistic and associated p-value are presented for each comparison[†].**

| Age | Total | Female | Male | Male vs Female | Female vs Male | χ² | *p*-value |
|---|---|---|---|---|---|---|---|
| **20-29** | 24.1% (23.4% - 24.8%) | 23.1% (21.9% - 24.2%) | 24.8% (23.8% - 25.7%) | 1.07 [1.01, 1.15] | 0.93 [0.87, 0.99] | 4.71 | 0.0299 |
| **30-39** | 27.0% (26.0% - 28.0%) | 24.7% (23.2% - 26.2%) | 28.7% (27.3% - 30.0%) | 1.16 [1.07, 1.25] | 0.86 [0.80, 0.93] | 13.97 | 0.0002 |
| **40-49** | 29.1% (27.6% - 30.5%) | 29.6% (27.3% - 31.8%) | 28.6% (26.7% - 30.6%) | 0.97 [0.87, 1.07] | 1.03 [0.93, 1.14] | 0.32 | 0.5724 |
| **50-59** | 35.7% (33.4% - 37.9%) | 39.3% (35.8% - 42.8%) | 33.0% (30.2% - 35.9%) | 0.84 [0.74, 0.95] | 1.19 [1.05, 1.35] | 7.12 | 0.0076 |
| **60+** | 45.9% (41.9% - 49.9%) | 51.1% (45.1% - 57.2%) | 41.7% (36.4% - 47.0%) | 0.82 [0.69, 0.97] | 1.23 [1.03, 1.46] | 4.86 | 0.0274 |
| **Total** | 26.9% (26.3% -27.4%) | 26.3% (25.5% - 27.1%) | 27.3% (26.6% - 28.0%) | 1.04 [1.00, 1.08] | 0.96 [0.92, 1.00] | 3.08 | 0.0793 |

[†]All data are derived from Special Olympics athletes ≥20 years old in the Special Olympics Healthy Athletes database.

adjusting the binomial logistic regression models for WHO regions, the prevalence of LBMD increased 1.78% [1.45%, 2.11%] (Z = 10.46, *p < .0001*) for males and 3.13% [2.74%, 3.53%] (Z = 15.37, *p < .0001*) for females for every year of age. The 20–29-year-olds had the lowest LBMD prevalence of 24.1% (23.4% - 24.8%) while the 60 + year-olds had the highest LBMD prevalence 45.9% (41.9% – 49.9%). When comparing by sex, males have higher prevalence rate ratios of LBMD than females at younger ages (20–29-year-olds: RR = 1.07, [1.01, 1.15]; 30–39 year olds: 1.16 [1.07, 1.25]) while 50–59 year old males (0.84 [0.74, 0.95]) and 60 + year old males (0.82 [0.69, 0.97]) have lower rates of LBMD than females of the same age (**Table 2**).

### Low bone mineral density - WHO Regions

The prevalence rate of LBMD varies by the WHO region. The Americas had the lowest prevalence rate (24.9% (24.3% - 25.5%)) while Southeast Asia had the highest prevalence rate (47.3% (44.0% - 50.5%)) (Fig 2). When normalized to

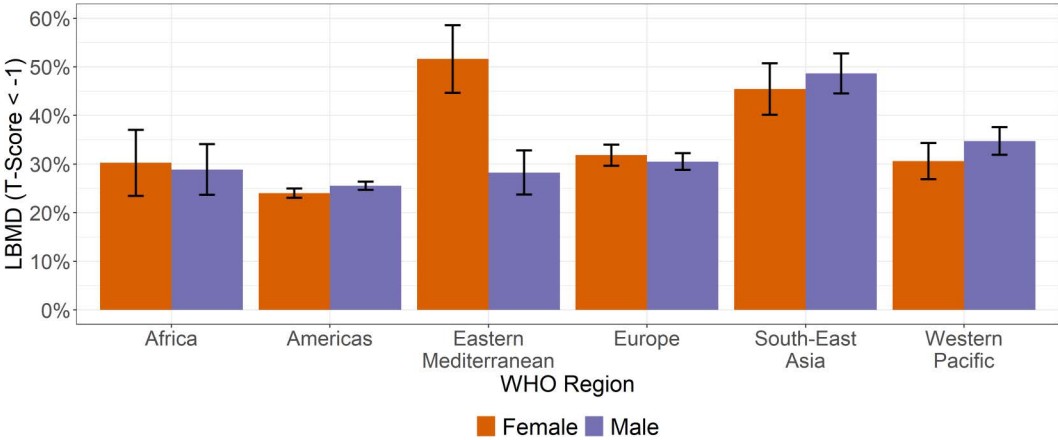

**Fig 2. The prevalence rate of LMBD for females and males across the WHO regions normalized to the Americas age distribution. Error bars represent 95% confidence intervals.**

the Americas Region, Eastern Mediterranean females had the highest rate of LBMD (51.6% (44.7% - 58.6%)) along with Southeast Asia males (48.6% (44.5% - 52.7%)) and females (45.4% (40.1% - 50.7%)). The Americas Region had the lowest rates of LBMD for both females (24.0% (23.1% - 25.0%)) and males (25.6% (24.7% - 26.4%)) (S1 Table).

Among the 20–29 age group, the regions and sexes with the highest prevalence of LBMD remained consistent, though their ranking varied slightly: 1) Southeast Asia females (44.8% (39.0% - 50.5%)); 2) Eastern Mediterranean females (37.2% (29.4% - 45.1%)); and 3) Southeast Asia males (35.8% (31.4% - 40.2%)). The lowest prevalence of LBMD was the African Region for both females (16.7% (10.4 - 22.9%)) and males (17.5% (12.5% - 22.5%)). For both the normalized population and the population restricted to 20–29-year-olds, Southeast Asia males had higher rates of LBMD compared to males in other regions. Southeast Asia and Eastern Mediterranean females had the highest prevalence by region (S2 Fig).

When the rates of LBMD among females were adjusted to match the male age distribution within each region, the rates for females and males were consistent, except for the Eastern Mediterranean. (Fig 3, S2 Table). There, the prevalence rate ratio of LBMD for females compared to males was 1.55 [1.24, 1.94] ($\chi^2$ = 13.52, $p$ = 0.0002). Eastern Mediterranean females had a rate of LBMD of 43.8% (36.9% - 50.7%), which is 15.5% higher than Eastern Mediterranean males, 28.3% (23.8% - 32.8%). In the Americas, the female-to-male prevalence rate ratio was 0.93 [0.88, 0.98] ($\chi^2$ = 7.08, $p$ = 0.0078). The LBMD rate among males in the Americas was 1.8% higher than that of females, with rates of 25.6% (24.7%–26.4%) and 23.8% (22.8%–24.8%), respectively.

## Peak bone mass

More than half of all athletes 54.9% (54.3% - 55.5%) failed to achieve a healthy PBM within the age range of 20–29 years with more males failing to achieve PBM 55.9% (55.1% - 56.7%) than females 53.4% (52.4% - 54.3%)*.* There is geographic variation in failing to achieve PBM with Southeast Asia females having the highest rates (75.9% (70.9% - 80.8%)) followed by Eastern Mediterranean females (66.9% (59.2% – 74.6%)) (Fig 4, S3 Table, S3 Fig). For males, Africa and Americas have the lowest rates of failing to achieve PBM compared to the other four regions.

## Discussion

The purpose of this study was to identify the overall prevalence of LBMD among adults with IDD and which groups were at greatest risk of poor bone health in terms of age groups, sex, and WHO global region. Our findings reveal that over a quarter of athletes screened at HA events had LBMD, with significant regional and sex differences. People with IDD have

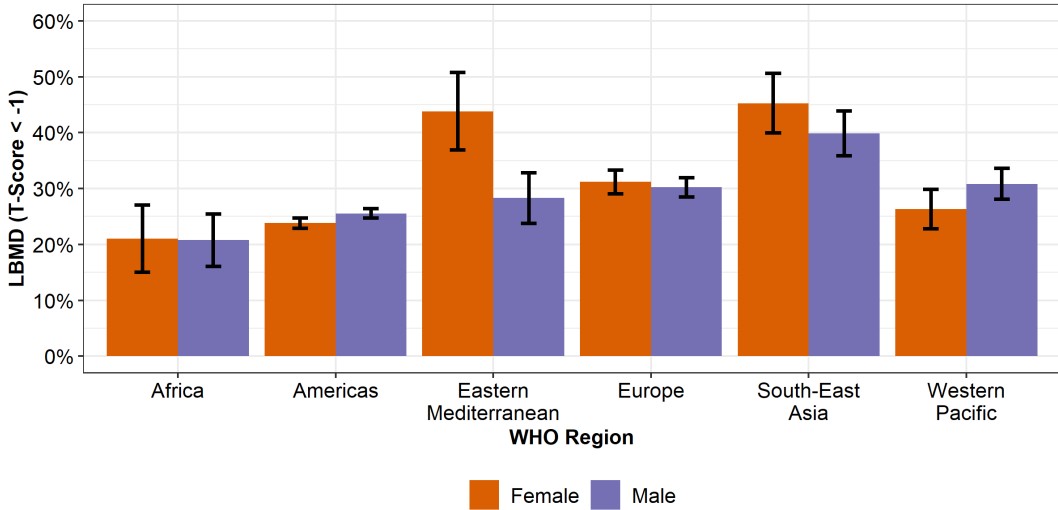

**Fig 3. Prevalence rates of LBMD by sex and WHO region where female age distribution in each region is normalized to their corresponding male age distribution.** Error bars represent 95% confidence intervals.

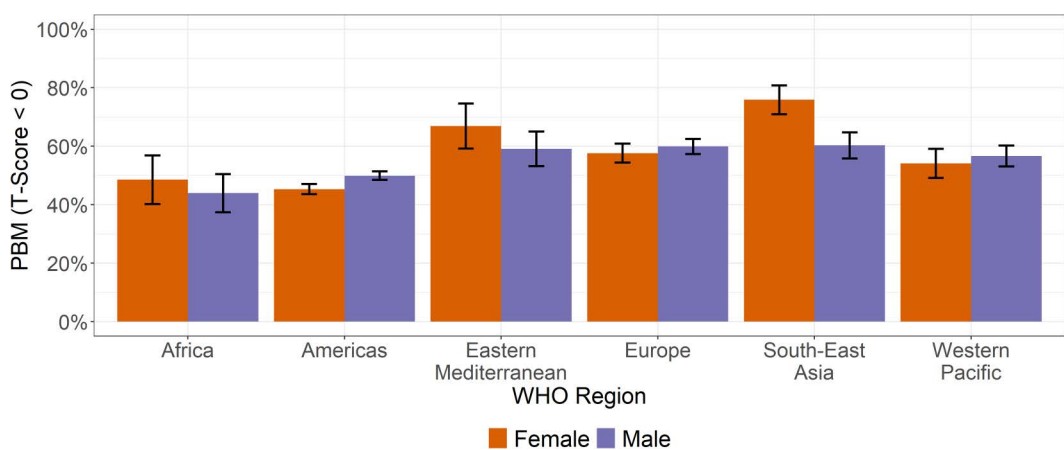

**Fig 4. Prevalence rates of failing to achieve a healthy PBM for 20–29-year-olds by sex and WHO region.** Error bars represent 95% confidence intervals.

higher rates of LBMD and fractures as well as earlier onset than the general population due to reduced mobility, lower physical activity, medication use, and nutritional deficiencies [14,27–29]. Notably, rates of LBMD increased with age, and males aged 20–39 had higher LBMD rates compared to females of the same age group. Other findings revealed that more than half of the 20–29-year-olds screened failed to achieve PBM. These results highlight variation in LBMD and failure to achieve PBM for the IDD community across age-groups, sex, and WHO regions.

Older athletes (50 + years) of both sexes had higher rates of LBMD, with female athletes being particularly affected. This trend aligns with patterns observed in the general population [30,31]. In females, LBMD is commonly associated with diminished estrogen levels following menopause [32], while males have age-related declines in testosterone and other factors contributing to bone loss [33]. An unexpected finding among 20–39-year-olds was that males had higher rates of LBMD than females. Further research is needed to determine whether this disparity reflects underlying physiological

differences, lifestyle factors, or measurement limitations. Targeted strategies to improve bone health across all age groups are essential, with particular emphasis on raising awareness about the risk of LBMD in younger adults with IDD, their caregivers, and the medical professionals who serve them.

This study identified regional differences in LBMD among adults with IDD. The Americas had the lowest rates, while South-East Asia exhibited the highest rates among males and among the highest rates for females. The elevated rates of LBMD in South-East Asia may be attributable to environmental or genetic influences [34,35]. Additionally, the Eastern Mediterranean region had the highest LBMD rates among females and was the only region where females had significantly higher rates than males. Certain lifestyle or environmental factors, such as limited sun exposure, may contribute to reduced vitamin D levels, potentially contributing to lower BMD in females from this region [36,37]. Further research is needed to understand regional variation in LBMD within the IDD population.

Achieving PBM is essential to help mitigate the risk of LBMD as people with IDD age. Failing to achieve PBM places people at an increased risk of hip fractures and, consequently, a reduced quality of life as they age [5]. Given that screenings for BMD are not recommended for young adults in the general population, heightened awareness of the LBMD risk for people with IDD will require targeted advocacy. For example, the U.S. Preventive Services Task Force (USPSTF) recommends BMD screening for women aged 65 years or older and for younger postmenopausal women who are at increased risk of osteoporosis [38]. For men, the USPSTF states that there is insufficient evidence to recommend routine screening, but men with a 10-year risk of osteoporotic fracture equal to or greater than that of a 65-year-old white woman without risk factors may benefit from screening. More needs to be done at the pre-screening age to help those with IDD achieve PBM and decrease their chance of LBMD or at least delay its onset.

Regional variation in failure to achieve PBM was observed, with sex differences playing a significant role. Among females, South-East Asia has the highest rates of failure to achieve PBM, possibly influenced by environmental or genetic factors [32,33]. Additionally, Eastern Mediterranean females have higher rates of failing to achieve PBM than all regions except South-East Asia, reflecting a similar pattern to LBMD rates. For males, the America and Africa regions exhibited lower rates of PBM failure compared to other regions, potentially due to physiological differences in BMD across race. Ettinger et al. found higher BMD in Black men compared to White men [39]. Our study lacked participant race and ethnicity information, making it difficult to confirm if racial differences contribute to regional differences. Further research is needed to explore why males with IDD in the Americas and Africa regions show better PBM outcomes than other regions.

Optimizing BMD in people with IDD requires a multifaceted approach, including proper nutrition, education, and opportunities for physical activity. Ensuring that children with IDD receive key nutrients to help increase BMD as they develop can help individuals achieve PBM. Calcium is a key component of bone structure [40,41] and increases in intake results in increases in BMD. Vitamin D is another important nutrient for BMD because it facilitates the absorption of calcium [41,42]. Physical activity is another critical factor in promoting bone health, as weight-bearing and resistance exercises stimulate bone formation and help maintain bone strength [43–45]. People with IDD face barriers to participating in regular physical activity, including limited access to structured programs, lack of community support, and lower physical literacy [43,44].

Special Olympics is uniquely positioned to support bone health in people with IDD through both sport participation and health programming. As a global sports organization, Special Olympics provides year-round training and competition in weight-bearing sports that can help improve BMD. Additionally, many outdoor sports offer natural opportunities for vitamin D production through sun exposure. Beyond sports, the Special Olympics integrates BMD screenings and health education through HA. While HA screenings provide valuable information, not all athletes have access to them, underscoring the need for expanded educational and healthcare initiatives.

Future research should also explore public health initiatives and policy strategies to address modifiable risk factors common among people with IDD including food insecurity [46], poor nutrition [47,48], lack of routine BMD screenings [14,29,49], vitamin D testing [40], limited weight-bearing physical activity [50], hormone imbalances [9], and side effects of

long-term medication use [51]. Priority should be given to research into applying interventions that support the development of a healthy PBM to promote lifelong bone health in this population.

This study has several limitations. Findings are from a sample of adults with IDD who participated in Special Olympics health programming and may not be generalizable to the broader population of persons with IDD. The study involves a convenient sample of Special Olympics athletes from 179 countries. However, the majority of athletes are from the Americas region, resulting in uneven global sampling. Nearly half the sample is between 20 and 29 years old. This uneven age sampling makes it difficult to compare rates of LBMD to general population rates, as it is uncommon for BMD screening in this age group. In addition, since longitudinal data does not exist for the Special Olympics athletes in this study, we were not able to account for confounding factors that impact BMD over time such as nutrition, activity levels and medication use. Nonetheless, this is the largest global study of BMD among people with IDD.

## Conclusion

This global study highlights the prevalence rates of LBMD for Special Olympics athletes with IDD across age-groups, sex, and WHO regions. The findings highlight regional disparities LBMD and PBM by WHO Global region, sex, and age. Preventive actions should begin in childhood and continue throughout life to optimize PBM, slow bone loss, and reduce the risk of fractures. Special Olympics plays a pivotal role in this space, offering opportunities for BMD screenings, physical activity, and health education that can positively impact BMD outcomes. Broader public health initiatives and policy changes are necessary to ensure people with IDD receive the support needed to develop and maintain healthy bones throughout their lives.

## Supporting information

**S1 Fig. Prevalence rate ratio heat maps of A) Females and B) Males by age group.** Each grid contains the prevalence rate ratio and [95% CI]. Colors gradients correlate with the magnitude of the prevalence rate ratios; note all 95% CIs that contain one have their tiles colored white.
(TIF)

**S2 Fig. LBMD prevalence rate ratio heat maps of the A) Females all ages normalized to Americas Region and B) Males all ages normalized by to Americas Region.** Each grid contains the prevalence rate ratio and [95% CI]. Colors gradients correlate with the magnitude of the prevalence rate ratios; note all 95% CIs that contain one have their tiles colored white.
(TIF)

**S3 Fig. Failure to achieve PBM prevalence rate ratio heat maps of the A) 20–29 year-old Females and B) 20–29 year-old Males.** Each grid contains the prevalence rate ratio and [95% CI]. Colors gradients correlate with the magnitude of the prevalence rate ratios; note all 95% CIs that contain one have their tiles colored white.
(TIF)

**S1 Table. Prevalence rates, with 95% confidence intervals, of LBMD for females and males.** The age distributions for each WHO region were normalized to the WHO Americas age distribution. See S2 Fig to compare the prevalence rate ratios of LBMD between WHO regions within a given sex.
(DOCX)

**S2 Table. Prevalence rates and prevalence rate ratios, with 95% confidence intervals, of LBMD where within each region the female age distribution is normalized to the male age distribution.** In addition, the Chi-square statistic and associated p-value are presented for each comparison.
(DOCX)

**S3 Table. Prevalence rates, with 95% confidence intervals, of failing to achieve PBM for 20–29-year-olds by sex and WHO region.** See S3 Fig to compare the prevalence rate ratios of LBMD between WHO regions within a given sex.
(DOCX)

**S1 Data. Data file.**
(CSV)

## Author contributions

**Conceptualization:** Mary Pittaway, John P. Hanley, Andrew E. Lincoln, Alicia M. Dixon-Ibarra, John T. Foley.

**Formal analysis:** John P. Hanley, Andrew E. Lincoln.

**Investigation:** Andrew E. Lincoln, Alicia M. Dixon-Ibarra.

**Methodology:** John P. Hanley, Andrew E. Lincoln, Alicia M. Dixon-Ibarra, John T. Foley.

**Project administration:** John P. Hanley, Andrew E. Lincoln, Alicia M. Dixon-Ibarra.

**Visualization:** John P. Hanley.

**Writing – original draft:** Mary Pittaway, John P. Hanley, Andrew E. Lincoln, Alicia M. Dixon-Ibarra, John T. Foley.

**Writing – review & editing:** Mary Pittaway, John P. Hanley, Andrew E. Lincoln, Alicia M. Dixon-Ibarra, John T. Foley.

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
