## [Decision Letter · Decision Letter 0]

30 May 2025

PGPH-D-24-03053

Bone Health Surveillance in Adults with Intellectual and Developmental Disability

Dear Dr. Foley,

Thank you for submitting your manuscript to PLOS Global Public Health. After careful consideration, we feel that it has merit but does not fully meet PLOS Global Public Health’s publication criteria as it currently stands. Therefore, we invite you to submit a revised version of the manuscript that addresses the points raised during the review process.

We look forward to receiving your revised manuscript.

Kind regards,

Javier H Eslava-Schmalbach, M.D., Ph.D., MSc

Academic Editor

Journal Requirements:

1. “Figure_2.tif, Figure_4.tif, Supplemental_Figure_2.tif and Supplemental_Figure_3.tif” files are over our file size limit of 10MB. This limit is in place for the convenience of reviewers, editors and readers. Please adapt this file so that the file size is below 10MB. You may find it helpful to consult our guidelines on compressing figures here: https://journals.plos.org/globalpublichealth/s/figures

Alternatively, you may wish to deposit large files in a separate repository to which you may link in your manuscript. You may find suggestions for repositories, including those which can accommodate large datasets, here: https://journals.plos.org/globalpublichealth/s/recommended-repositories

2. We do not publish any copyright or trademark symbols that usually accompany proprietary names, eg (R), (C), or TM (e.g. next to drug or reagent names). Please remove all instances of trademark/copyright symbols throughout the text, including ® on pages 2, 4, 9, 11 and 12.

Additional Editor Comments (if provided):

Dear authors:

We have received reviewers' comments. Please comment and include reviewers' suggestion in the article. Also, I am including my suggestions:

Title:Please include the study design in the title, as suggested by STROBE.

Study Population and Data:In line 89, it is mentioned that this is a secondary data study, and in line 114, you state “there were 25,868 athletes ≥ 20 years old from 179 countries who received BMD testing.” In Methods section, it is important to clarify the universe size (N) and the number of athletes in the universe who received BMD testing (n), so readers will understand how representative the sample is. Additionally, please explain possible reasons why some athletes were not included.

Gender/Sex Variable:The female-male variable is mentioned in the paper. However, it is not clear whether athletes of other genders were excluded from the analysis, not accepted in the competitions, or if you decided to use biological sex for the analysis or to exclude other genders. Please clarify this in the methods. This should be clarified

Statistical Analysis:

Line 109: Binomial logistic regression is mentioned. It appears to be a crude binomial logistic regression with age as a continuous variable, even though region and the female-male variable are confounders in this relationship. I suggest running both crude and adjusted logistic regression in this case.

Lines 139–140: Age was managed as a string variable. Please clarify in the methods section why two types of age analysis were used, or consider unifying them.

Reference:Line 116: There is a reference not inserted.

Tables and Figures:

Table 1: Please include p-values for the observed differences in these variables.

General: Add the source of each table and figure as a footnote.

Line 138: Please change “rate ratio” to “prevalence rate ratio” in this line and elsewhere in the text where “prevalence” is missing (e.g., line 169).

Line 152: I suggest reviewing this text. The prevalence rate of LBMD (the outcome) is being presented, not the prevalence rates of males and females.

Figures:

Line 188: Figure 4 was attached as Figure 3. Please review this.

Supplementary Figure 1: The heat map of prevalence rate ratio shows redundant information when comparing the risk of LBMD (e.g., comparing 20–29 years to 60–69 years, then 60–69 years to 20–29 years). I suggest revising this figure to exclude redundant information and, for example, compare 60–69 years males prevalence rate ratios to 60–69 years females prevalence rate ratios, and so on for other categories.

Supplementary Figure 2: The same comment as for Supplementary Figure 1 applies.

Supplementary Figure 3: The same comment as for Supplementary Figure 1 applies.

Supplemental Table 4: This table should be sorted as follows: second column—Total, Male, Female, Male vs Female Ratio, p-value (from Table 2). Do not use scientific notation for p-values (e.g., 2.99 × 10⁻²); use standard notation (e.g., 0.0290, four decimals). This suggestion applies to all tables and figures in the article. For p-values less than 0.0001, note as <0.0001.

Supplemental Table 2: With the previous changes, this table is not needed.

Supplemental Table 3: This table should be sorted as follows: second column—Total, Male, Female, Male vs Female Ratio, p-value. Review the title; it refers to the prevalence of LBMD (the outcome), not the prevalence of females or males.

Supplemental Table 4: This table should be sorted as follows: second column—Total, Male, Female, Male vs Female Ratio, p-value (from Table 5).

Supplemental Table 5: With the previous changes, this table is not needed.

Supplemental Table 6: This table should be sorted as follows: second column—Total, Male, Female, Male vs Female Ratio, p-value.

Reviewers' comments:

Reviewer's Responses to Questions

**Comments to the Author**

1. Does this manuscript meet PLOS Global Public Health’s publication criteria?

Reviewer #1: Yes

Reviewer #2: Yes

Reviewer #3: Partly

Reviewer #4: Yes

Reviewer #5: Yes

2. Has the statistical analysis been performed appropriately and rigorously?

Reviewer #1: Yes

Reviewer #2: Yes

Reviewer #3: No

Reviewer #4: Yes

Reviewer #5: Yes

3. Have the authors made all data underlying the findings in their manuscript fully available (please refer to the Data Availability Statement at the start of the manuscript PDF file)?

Reviewer #1: Yes

Reviewer #2: Yes

Reviewer #3: Yes

Reviewer #4: Yes

Reviewer #5: Yes

4. Is the manuscript presented in an intelligible fashion and written in standard English?

Reviewer #1: Yes

Reviewer #2: Yes

Reviewer #3: Yes

Reviewer #4: Yes

Reviewer #5: Yes

Reviewer #1: The manuscript was well written. The authors had conducted a cross sectional study using secondary data to assess the prevalence of low bone mineral density among adults with intellectual disability participating in special olympics. The research is relevant and reports the adverse consequences of LBMD in adults. The authors acknowledge the current gaps in research on prevalence of LBMD in adults with IDD. Their methodology and the their statistical analysis described the sociodemographic variables and prevalence rates and the went further to conduct a regression analysis in line with the variables in the data. In the discussion section when comparing study findings with other research, it would be useful to include more details of the other research such as if the study was conducted in a similar population or among individuals with IDD.

Reviewer #2: Dear Authors,

I have provided a detailed feedback in an attachment. Overall, this is a well-written and timely article that addresses an important and underexplored area of health among individuals with intellectual and developmental disabilities. The authors are to be commended for their comprehensive analysis and thoughtful discussion, well done.

Reviewer #3: Title – Kindly include “Athletes with IDD” to give clarity on the population of interest

Methods -

• Is there a possibility of duplication of data of participants who appeared more than once to the Olympics? If yes, which values were included and excluded?

• Are BMD T scores comparable across geographical regions? Was cross region adjustments and standardization done before interpretation? Kindly add reference

• Sensitivity and specificity of the Quantitative USG used? Kindly add reference

• Can WHO classification of osteoporosis thresholds apply to QUS results? Kindly add reference

• Implications of restricting assessment only to a single site (heel)

• Were appropriate confounders adjusted for? Kindly mention the variables adjusted for in analysis.

• Scope of adding more relevant variables in the regression model – physical activity, BMI, nutrition, medication history, etc

Results

• Add atleast the supplementary table 1 and 2 (containing prevalence of LBMD) in the main article after combining them in the same table. As this is your primary objective, these results should feature in the main article

• Can we make meaningful comparison of different regions when the disparity in sample sizes is significant?

Discussion

• Need to mention the implications of all possible selection and measurement bias in detail in discussion

Reviewer #4: Overall this is an excellent retrospective cohort study assessing rates of Low Bone Mineral Density among Special Olympics athletes with intellectual or developmental disability.

Notable findings include the low rates of achieving peak bone mineral density in 20-29yo, the differences in gender prevalence of LBMD at different age groups, and the remarkable difference in incidence based on geographic location.

It would be helpful for the authors to address the following prior to publication:

- please comment upon the reliability of using US for BMD screening. How does this compare to DEXA so that we know how much to accept these results?

- At several points, there were comments upon the increasing % of risk of LBMD per year of age. Did the study team actually plot out the rates by year of age or by decile of age? If they did assess each year of age individually, it would be really nice to see that data in the supplemental figures.

- The authors did a nice job of introducing some initial ideas around LBMD prevention within this population. Do they have any specific recommendations for the younger populations with LBMD with regards to screening and prevention that we should perhaps include?

Reviewer #5: My comments:

1. The manuscript addresses an underrepresented population (individuals with intellectual and developmental disabilities, IDD) in the context of bone health, an area of significant public health concern.

2. The recommendations for public health interventions are solid but could be improved by adding specific examples or strategies suited for low-resource settings.

**Do you want your identity to be public for this peer review?** For information about this choice, including consent withdrawal, please see our Privacy Policy

Reviewer #1: No

Reviewer #2: No

Reviewer #3: No

Reviewer #4: No

Reviewer #5: **Yes: ** HAFASHIMANA Valens

---

## [Decision Letter · Decision Letter 1]

8 Aug 2025

Global Variation of Low Bone Mineral Density in Special Olympics Adult Athletes with Intellectual and Developmental Disability - A Cross-sectional Study.

PGPH-D-24-03053R1

Dear Prof Foley,

We are pleased to inform you that your manuscript 'Global Variation of Low Bone Mineral Density in Special Olympics Adult Athletes with Intellectual and Developmental Disability - A Cross-sectional Study.' has been provisionally accepted for publication in PLOS Global Public Health.

Best regards,

Julia Robinson

Executive Editor

Reviewer Comments (if any, and for reference):

Reviewer's Responses to Questions

**Comments to the Author**

Reviewer #1: All comments have been addressed

Reviewer #2: All comments have been addressed

Reviewer #3: All comments have been addressed

publication criteria?

Reviewer #1: Yes

Reviewer #2: Yes

Reviewer #3: Yes

3. Has the statistical analysis been performed appropriately and rigorously?

Reviewer #1: Yes

Reviewer #2: Yes

Reviewer #3: Yes

4. Have the authors made all data underlying the findings in their manuscript fully available (please refer to the Data Availability Statement at the start of the manuscript PDF file)?

Reviewer #1: Yes

Reviewer #2: Yes

Reviewer #3: No

5. Is the manuscript presented in an intelligible fashion and written in standard English?

Reviewer #1: Yes

Reviewer #2: Yes

Reviewer #3: Yes

Reviewer #1: The authors have addressed my previous comments. The only observed comment I have is to correct the in text citation error in the results section seen last line before table 1.

Reviewer #2: Thank you for addressing the issues raised, well done.

Reviewer #3: (No Response)

**Do you want your identity to be public for this peer review?** For information about this choice, including consent withdrawal, please see our Privacy Policy

Reviewer #1: No

Reviewer #2: No

Reviewer #3: No
